# Impact of diabetes self-management, diabetes management self-efficacy and diabetes knowledge on glycemic control in people with Type 2 Diabetes (T2D): A multi-center study in Thailand

**Cameron P. Hurst**[1,2☯]**, Nitchamon Rakkapao**[2☯]*****, Karen Hay**[1]

**1** QIMR Berghofer Medical Research Institute, Brisbane, Queensland, Australia, **2** Faculty of Public Health, Thammasat University, Muang, Lampang, Thailand

☯ These authors contributed equally to this work.
* nitchamonbt@fph.tu.ac.th

**Data Availability Statement:** A de-identified version of our data along with a data dictionary are available from the Mendeley data repository (doi:

## Abstract

We investigate the relationship of diabetes knowledge, diabetes management self-efficacy and diabetes self-management with blood glucose control among people with Thai type 2 diabetes mellitus (T2D). Seven hundred outpatients from diabetes clinics from large university and small community hospitals in two provinces of Thailand (Khon Kaen and Bangkok) were interviewed to evaluate their diabetes knowledge (DK), diabetes management self-efficacy (DMSE) and diabetes self-management (DSM). In addition, patient medical records were accessed to obtain other patient characteristics including patients' HbA1c levels. Bivariate and multivariable logistic regression modelling was conducted and unadjusted and adjusted odds ratios obtained, respectively. Over half (52.4%) of the patients in our sample failed to control their blood glucose (HbA1c > 7%). All three psychometric measures (DK, DMSE and DSM) were identified as associated with blood glucose control in the bivariate analysis ($OR_{DK(unadj)}$ = 0.89, 95%CI: 0.82, 0.96; $OR_{DSM(unadj)}$ = 1.64, 95%CI: 1.46, 1.82; $OR_{DMSE(unadj)}$ = 2.84; 95%CI: 2.43, 3.32). However, after mutual adjustment and adjustment for other patient characteristics, of the three psychometric measures, only diabetes management self-efficacy remained associated with blood glucose control ($OR_{DMSE(adj)}$ = 2.67; 95%CI: 2.20, 3.25). Diabetes management self-efficacy is shown to be strongly associated with blood glucose control in the Thai Type 2 diabetes population. Current early diabetes interventions in Thailand tend to focus on disease knowledge. A stronger emphasis on enhancing patients' disease management self-efficacy in these interventions is likely to lead to substantial improvement in both diabetes self-management and blood glucose control, thereafter reducing the risk, or prolonging the development, of chronic diabetes complications.

10.17632/xhmjcfp2ym.1): https://data.mendeley.com/datasets/xhmjcfp2ym/1.

**Funding:** The authors received no specific funding for this work.

**Competing interests:** The authors have declared that no competing interests exist.

## Introduction

Diabetes mellitus is a major health problem and represents a substantial burden in terms of mortality, morbidity, and health-system costs, worldwide. In 2015, the number of people living with diabetes was estimated to be 415 million globally and this number is expected to rise to 642 million cases by 2040 [1]. Approximately 90% of people with diabetes have Type 2 Diabetes Mellitus (T2D) and the increasing prevalence of T2D is likely to be attributable to factors such as ageing populations and an increasing level of sedentary life styles that seem to accompany economic development. Importantly, over 80% of T2D cases live in low and middle-income countries [2].

Thailand has recently experienced a major increase in the prevalence of T2D, increasing from 6.9 to 8.9% over the six year period 2009–2014 [3] and this is expected to rise to 9.8% by 2030 [4]. Unhealthy lifestyles due to urbanization, less physical activity and low consumption of fruits and vegetables, as well as high consumption of sugar are the main reasons for this increasing prevalence [5]. The disease burden is estimated to represent 21% per capita GDP [6], that is, approximately one fifth of an average Thai's economic output. The increasing prevalence of T2D, along with the associated development of T2D complications as a cause of early morbidity and mortality, and the enormous burden on health care systems make diabetes a priority health concern. In 2017, diabetes was one of the highest contributors to disability, ranked 5[th] in both years lived with disability (YLD) and disability-adjusted life years (DALYs), with increases of more than 50% over the 10 years period from 2007–2017 [7].

Glycated hemoglobin (HbA1c) level is an indicator of glycemic control in the 2–3 months prior to the test, with abnormally high HbA1c levels (HbA1c > 7%) having been shown to be associated with a higher risk of the development of complications and mortality [8]. Moreover, maintaining HbA1c control (HbA1c ≤ 7%) has been demonstrated as protective against the development of chronic T2DM complications such as diabetic retinopathy, diabetic nephropathy and diabetic neuropathy [9]. Achieving glycemic control has been shown to be strongly linked to effective diabetes self-management, or diabetes self-care, which comprises of good adherence to medication regimens, ongoing monitoring of diet and blood glucose levels, engaging in physical activity and effective foot care. Several studies have shown that improving diabetes self-management is important in achieving better health outcomes and reduced incidence of complications [10–12]. Indeed McDowell and Colleagues [13] suggest that strong diabetes self-management may be almost as efficacious as oral hypoglycemic agents in controlling blood glucose, especially in the earlier stages of the disease.

Effective diabetes self-management has been strongly linked to the antecedent constructs: diabetes management self-efficacy and diabetes knowledge. Self-efficacy is a person's confidence in their ability to perform a goal-directed behavior [14], and diabetes management self-efficacy, a diabetes patient's confidence in managing their own disease, has been demonstrated to be associated with better self-care behavior and glycemic control in patients with diabetes [15]. Diabetes knowledge traditionally emphasizes positive change through improving patient's knowledge of their disease. A large body of literature exists on diabetes knowledge and its effectiveness in improving diabetes clinical targets [16, 17].

Despite there being strong evidence on the link between diabetes management and glycemic control, several studies have demonstrated that many patients exhibit poor diabetes self-management (for example, [12, 18]). It is likely that factors associated with poor diabetes self-management, self-efficacy and knowledge and the complex interplay between these three constructs are likely to drive poor glycemic control. The objective of this study was to explore impact of diabetes self-management, diabetes management self-efficacy and diabetes knowledge on glycemic control in Thais with T2D with psychometric validated instruments.

## Materials and methods

### Study design and sample

This cross-sectional study included 700 people with T2D living in either rural or urban areas from the central and north-eastern regions of Thailand. Patients were recruited from outpatient diabetes clinics of both community and university hospitals in the Khon Kaen and Bangkok provinces of Thailand. The four hospitals from which participants were enrolled were the Phupaman community hospital in Khon Kaen, Srinagarind hospital at Khon Kaen University, and Wechkaroonrasm community hospital and the King Chulalongkorn Memorial Hospital (Chulalongkorn University) which are both located in Bangkok. As this was an observational study with three study effects (Diabetes Knowledge, management self-efficacy and self-management) and a large number of potential covariates we used the 10 cases per predictor approach advocated by Harell and colleagues (1996) [19] to estimate our sample size. Assuming a prevalence of blood glucose control of 0.36 [20] and up to 25 parameters in the final model, we estimated sample size of 695 patients (rounded up to 700) was required. Patients were sampled using a stratified sampling design with strata sizes based on locality (Province) —hospital size combinations. Questionnaires were administered in February to June, 2016 to T2D outpatients aged at least 20 years old who had had a diagnosis of T2D for at least 3 years, able to read and understand the Thai language and were willing to participate in the study. The authorized person of each hospital gave permission to collect the data, and all participants provided written informed consent. Potential participants were identified by the receptionist nurse upon arrival for their routine diabetes check-up. These patients were approached while waiting for their appointment and asked if they would participate. Of all of the patients approached (708) all agreed to participate, but eight patients were excluded as they were not fluent in the Thai language. The study protocol was approved by the ethics committee of Khon Kaen University (HE581479), the Institutional Review Board at the Faculty of Medicine, Chulalongkorn University (IRB035/59), and the Bangkok Metropolitan Administration Ethics Committee for Human Research (U005q/59). The English and Thai versions of the questionnaire are provided in S1 and S2 Files, respectively.

### Measurement of diabetes self-management, management self-efficacy and knowledge

We used three instruments to measure Diabetes self-management, Diabetes management self-efficacy, and diabetes knowledge. Diabetes self-management was measured using the Summary of Diabetes Self-Care Activities (SDSCA), a 17-item instrument originally developed by Toobert and colleagues [21]. Diabetes management self-efficacy was measured using the 20-item Diabetes Management Self-efficacy Scale (DMSES) designed by Van Der Bijl and colleagues [22]. Finally, diabetes knowledge was measured using a 10-item version of the Diabetes Knowledge Scales (DK) developed by Beeney and colleagues [23]. It should be noted that the original version of the Diabetes Knowledge scale included 17 items, but we removed all items specific to Type 1 diabetes or insulin treatment as the present study only considered people with T2D, many of whom were not using insulin treatment for glycemic control. All three instruments were originally developed in English and in a resource-sufficient health care setting. Consequently, our study conducted a validation of the instruments in the Thai health setting. This validation process was conducted in two phases: (1) Early phase validation which involved establishing translational and face validity, and (2) Demonstrating construct validity in Thai people with T2D. Permission to employ the DMSES, SDSCA and DK instruments was obtained from their respective authors [21–23].

**Instrument translation and face validity.** The items of all three instruments (SDSCA, DMSES and DK) were translated from English into Thai using the forward and backward translation technique outlined by Brislin [24]. Four Thai-English bilingual translators were identified, and of these, two were used to forward translate the original version of the items of the three instruments into Thai, while the remaining two translators were used to independently back-translate the items from Thai back to English. The original and back translated versions of the three instruments' items were then compared by two native English speakers. Finally the three instruments were field tested in a pilot group consisting of 20 people with T2D to evaluate the translational quality and the practical aspects of test administration. Participants were asked to read and listen to each item in order to ensure their understanding.

**Construct validity of SDSCA, DMSES and DK.** All items of the instruments for diabetes self-management (SDSCA) and diabetes management self-efficacy (DMSES) were measured on an ordinal scale. Most items in SDSCA scale relate to the frequencies of diabetes self-management activities in the previous week and are measured on an eight-point scale which represent the numbers of days in the previous week (0, 1, 2,. . .,7 days) the activity was performed. In the original form of the instrument there are also binary (about smoking) and nominal items (about doctor treatment recommendations). We removed both of these items. For the DMSES instrument all items are measured on a five point scale ranging from 1 = least confident to 5 = most confident. As both the SDSCA and DMSES instruments involve ordinal scale items we used exploratory and confirmatory factor analysis to elucidate and confirm each instrument's structure, respectively. Exploratory factor analysis was conducted using Principal Components Analysis followed by a Parallel analysis to determine the number of domains in the Thai T2D population. Principal Axis Factoring was subsequently performed to examine the nature of the domains in this population. Unweighted least squares Confirmatory Factor Analysis was then conducted on the resulting measurement models to determine whether they sufficiently fit our sample. We used the Cumulative-fit index (CFI), adjusted goodness-of-fit index (AGFI), root-mean-square error of approximation (RMSEA), and the Tucker-Lewis index (TLI) to gauge model fit. A model with TLI, CFI, GFI and AGFI > 0.9 [25, 26, 27, respectively], and RMSEA < 0.08 [28] was deemed to represent adequate model fit. Despite being well established as a poor measure of measurement model fit, we also included the $\chi^2$ goodness of fit statistic for reasons of convention. Bartlett's test of sphericity and the Kaiser-Meyer-Olkin (KMO) measure of sampling adequacy were generated along with the CFA to provide further evidence of construct validity [29]. Internal consistency reliability for both SDSCA and DMSES was assessed using Cronbach's alpha with $\alpha > 0.7$ considered adequate reliability [30]. To avoid model overfitting, the exploratory and confirmatory phases of the factor analyses were performed on a random split of the sample containing 200 and 500 of the total 700 patients, respectively.

As the DK instrument measures knowledge of diabetes, its items are binary (correct or incorrect answers). Consequently, standard factor analytic techniques which are designed for quantitative items are inappropriate. Instead, construct validity of the DK was examined using Multivariate Item Response Theory (MIRT) with items assumed to have a two parameter model with no guessing (2PL). A model with a Root mean square error of approximation (RMSEA) less than 0.05, and Tucker Lewis Index (TLI) and Cumulative Fit Index (CFI) exceeding 0.9 [31, 32] was deemed to have adequate fit. Internal consistency reliability of the DK was assessed using the Kuder-Richardson 20 index (KR20) with a model with KR20>0.7 assumed to be sufficiently reliable [33].

The DK instrument is represented by a single domain, while both the SDSCA and DMSES are multidimensional with both instruments being represented by four domains: Diet, Physical activity, Blood monitoring and Foot-care for SDSCA; and Diet SE, Physical activity SE,

Regimen SE and Monitoring SE for DMSE. Although two of the three instruments are multidimensional, we only employed the total scales in our logistic regression modelling (sums of the scores from the individual domains). Finally, to aid in comparing each instrument's effect sizes, we rescaled each instrument's total score onto a 10 unit scale (i.e. All three instruments yielded scores with a minimum of 0 and a maximum of 10) to give comparable units for the measures of diabetes self-management (SDSCA10), diabetes management self-efficacy (DMSES10) and diabetes knowledge (DK10).

## Other clinical and demographic variables

In addition to SDSCA, DMSES, and DK, the questionnaire included questions relating to socio-demographics including gender, marital status, age, education, religion, household income, family history of T2DM, smoking, and alcohol consumption. Also, clinical data such as comorbidities, duration of diabetes, type of diabetes treatment, weight, height and glycated hemoglobin (HbA1c) were extracted from each patient's electronic medical records using the current visit for each participant.

## Statistical analysis

Baseline patient characteristics were summarized using means and standard deviations for continuous variables, and counts and percentages for categorical variables. As, the outcome variable in this study, Glycemic control (No: HbA1c > 7%; Yes: HbA1c $\leq$7%) is binary, and this was a multi-centre study with a potential centre-clustering effect, binary logistic mixed effect regression modelling was employed to obtain all unadjusted and adjusted odds ratios. Potentially important risk factors and/or confounders were identified *a priori* based on literature review. To investigate the interplay among SDSCA10, DMSES10 and DK10 in explaining HbA1c control, we fit four multivariable models: (Model 1) SDSCA10 (alone) adjusted for patient characteristics (demographic and clinical); (Model 2a) SDSCA10 and DMSES10 adjusted for patient characteristics; (Model 2b) SCDACA10 and DK10 adjusted for patient characteristics; and (Model 3) SDSCA10, DMSES10 and DK10 adjusted for patient characteristics. All analysis was conducted in R (v3.2.0) [34], with factor analysis and multidimensional item response theory being performed using the lavaan [35] and mirt [36] R libraries, respectively. Binary logistic mixed effect regression modelling was conducted using the R library, lme4 [37]. A p-value less than 0.05 was used to gauge statistical significance throughout all inferential analysis.

## Results

### Patient characteristics

Table 1 provides the characteristic for all patients, and stratified by glycemic control status. In total, 700 T2DM outpatients were included in this study with a mean age 65.16 (SD = 10.94). A majority of the sample were female (n = 492, 70.3%), married (n = 462, 66%), had a primary school level of education (n = 381, 54.4%). Over half the patients in our sample (n = 367, 52.4%) had uncontrolled HbA1c.

 **Factors associated with glycemic control.** The unadjusted and adjusted odds ratios for factors potentially associated with glycemic control are provided in Tables 2 and 3, respectively. The bivariate analysis (Table 2) demonstrates that both diabetes self-management (ORSDSCA:10 = 1.62; 95%CI:1.46, 1.80; p < 0.001) and diabetes management self-efficacy (ORDMSES:10 = 2.20; 95%CI:1.97, 2.46 p < 0.001) were positively associated with glycemic control, while an inverse relationship was observed between diabetes knowledge

**Table 1. Patient characteristics.**

| Variable | HbA1c > 7 | HbA1c ≤ 7 | P-value | Total |
|---|---|---|---|---|
| **Number of patients** | 367 | 333 | | 700 |
| **SDSCA.10** (mean (sd)) | 4.20(1.59) | 5.47(1.64) | <0.001 | 4.80(1.73) |
| **DMSES.10** (mean (sd)) | 4.44(1.81) | 7.50(1.76) | <0.001 | |
| **DK.10** (mean (sd)) | 5.68(1.93) | 5.38(2.07) | 0.051 | 5.53(2.00) |
| **Sex**: Female (%) | 254(69.2) | 238(71.5) | 0.568 | 492(70.3) |
| **Age** (mean (sd)) | 63.13(10.83) | 67.40(10.63) | <0.001 | 65.16(10.94) |
| **Marital status** (%) | | | 0.003 | |
| Single | 21(5.7) | 36(10.8) | | 57(8.1) |
| Married | 262(71.4) | 200(60.1) | | 462(66.0) |
| WDS | 84(22.9) | 97(29.1) | | 181(25.9) |
| **Education** (%) | | | 0.030 | |
| No formal | 15(4.1) | 32(9.6) | | 47(6.7) |
| Primary | 209(56.9) | 172(51.7) | | 381(54.4) |
| Secondary | 78(21.3) | 68(20.4) | | 1.46(20.9) |
| Bachelors+ | 65(17.7) | 61(18.3) | | 126(18.0) |
| **Monthly Income** (%) | | | 0.033 | |
| <5K (<150USD) | 147(40.1) | 171(51.4) | | 318(45.4) |
| 5–9.99K (151-300USD) | 55(15.0) | 40(12.0) | | 95(13.6) |
| 10–14.99K (301-450USD) | 54(14.7) | 32(9.6) | | 86(13.2) |
| 15–24.99KK (451-750USD) | 54(14.7) | 42(12.6) | | 96(13.7) |
| > = 25+K (> = 751USD) | 57(15.5) | 48(14.4) | | 105(15.0) |
| **Province: KK** (%) | 93(25.3) | 44(13.2) | <0.001 | 137(19.6) |
| **Religion: Non Buddhist** (%) | 85(23.2) | 72(21.6) | 0.691 | 157(22.4) |
| **DM duration** (mean (sd)) | 13.65(7.95) | 13.41(8.77) | 0.701 | 13.53(8.34) |
| **DM treatment** (%) | | | <0.001 | |
| None | 3(0.8) | 9(2.7) | | 12(1.7) |
| OHA | 162(44.1) | 247(74.2) | | 409(58.4) |
| Insulin | 66(18.0) | 28(8.4) | | 94(13.4) |
| OHA+Insulin | 136(37.1) | 49(14.7) | | 185(26.4) |
| **Smoking** (%) | | | 0.680 | |
| Never | 311(84.7) | 278(83.5) | | 589(6.7) |
| Previous | 46(12.5) | 42(12.6) | | 88(12.6) |
| Current | 10(2.7) | 13(3.9) | | 23(3.3) |
| **Alcohol** (%) | | | 0.687 | |
| Never | 294(80.1) | 275(82.6) | | 569(81.3) |
| Previous | 49(13.4) | 40(12.0) | | 89(12.7) |
| Current | 24(6.5) | 18(5.4) | | 42(6.0) |
| **BMI class** (%) | | | 0.350 | |
| <18.5 | 11(3.0) | 9(2.7) | | 20(2.9) |
| 18.5–24.9 | 136(37.1) | 145(43.5) | | 281(40.1) |
| 25–29.9 | 122(33.2) | 95(28.5) | | 217(31.0) |
| ≥30 | 98(26.7) | 84(25.2) | | 182(260) |
| **Comorbidities** (%) | 341(92.9) | 317(95.2) | 0.267 | 658(94.0) |
| **DM Family history** (%) | 180(49.0) | 150(45.0) | 0.325 | 330(47.1) |
| **Hospital size: small** (%) | 172(46.9) | 129(38.7) | 0.036 | 301(43.0) |

p-values from t-test for continuous variables, and chi-square tests of independence for categorical variables.

**Table 2. Unadjusted odds ratios for diabets self management, diabetes management self efficacy, diabetes knowledge and other patient characteristics for the outcome, HbAIc control.**

| Effect | OR | 95% CI |
|---|---|---|
| **SDSCA10** | 1.634*** | 1.463, 1.824 |
| **DMSES10** | 2.836*** | 2.425, 3.316 |
| **DK10** | 0.886** | 0.816, 0.961 |
| **Sex**: Female | 1.120 | 0.802, 1.564 |
| **Age** (10 years) | 1.394*** | 1.190, 1.634 |
| **Marital status (Ref: Single)** | $\chi^2 = 7.563$, df = 2, p = 0.022 | |
| Married | 0.525* | 0.293, 0.939 |
| WDS | 0.756 | 0.406, 1.405 |
| **Education (Ref: No formal)** | $\chi^2 = 7.289$, df = 3, p = 0.063 | |
| Primary | 0.436* | 0.226, 0.842 |
| Secondary | 0.406* | 0.201, 0.823 |
| Bachelors+ | 0.433* | 0.210, 0.892 |
| **Income (Ref: <5K)** | $\chi^2 = 6.752$, df = 4, p = 0.150 | |
| 5–9.99K (151-300USD) | 0.749 | 0.461, 1.216 |
| 10–14.99K (301-450USD) | 0.573* | 0.347, 0.949 |
| 15–24.99KK (451-750USD) | 0.681 | 0.427, 1.087 |
| 25+K (>750USD) | 0.708 | 0.447, 1.122 |
| **Province**: KK | 0.452** | 0.289, 0.709 |
| **Religion**: Non Buddhist | 0.971 | 0.612, 1.540 |
| **DM duration** (5 years) | 0.917 | 0.832, 1.011 |
| **DM treatment (Ref: None)** | $\chi^2 = 82.277$, df = 3, p<0.001 | |
| OHA | 0.670 | 0.177, 2.542 |
| Insulin | 0.166* | 0.042, 0.664 |
| OHA+Insulin | 0.146** | 0.038, 0.568 |
| **Smoking (Ref: Never)** | $\chi^2 = 1.375$, df = 2, p = 0.505 | |
| Previous | 1.105 | 0.693, 1.761 |
| Current | 1.630 | 0.692, 3.838 |
| **Alcohol (Ref: Never)** | $\chi^2 = 0.114$, df = 2, p = 0.945 | |
| Previous | 0.970 | 0.605, 1.555 |
| Current | 0.897 | 0.467, 1.723 |
| **BMI class (Ref: 18.5–24.9)** | $\chi^2 = 3.669$, df = 3, p = 0.300 | |
| <18.5 | 0.793 | 0.314, 2.001 |
| 25–29.9 | 0.706 | 0.491, 1.014 |
| 30+ | 0.810 | 0.548, 1.197 |
| **Comorbility: Yes** | 1.424 | 0.737, 2.753 |
| **Family history: Yes** | 0.833 | 0.616, 1.127 |
| **Hospital size: Small** | 0.760 | 0.353, 1.638 |

Note: $\chi^2$ represents the Likelihood ratio test;

***P<0.001,

**P<0.01,

*P<0.05

(ORDK10 = 0.89; 95%CI:0.82, 0.96; p = 0.003) and glycemic control. A positive association was observed between age and HbA1c control with each 10 years of age being associated with a 39% increase in the odds of glycaemic control (ORAge = 1.394; 95%CI: 1.19, 1.634; p < 0.001). Married patients (relative to single patients), Khon Kaen patients (relative to

**Table 3. Four different models of blood-glucose control in terms of different combinations of the diabetes knowledge, management self-efficacy and self-management predictors.**

| Effect | $OR_{Model1}$ | $OR_{Model2a}$ | $OR_{Model2b}$ | $OR_{Model3}$ |
|---|---|---|---|---|
| **SDSCA10** | 1.69(1.49, 1.91)*** | 1.11(0.95, 1.29) | 1.69(1.50, 1.91)*** | 1.11(0.95, 1.29) |
| **DMSE10** | - | 2.68(2.20, 3.26)*** | - | 2.67(2.20, 3.25)*** |
| **DK10** | - | - | 0.94(0.83, 1.06) | 0.96(0.84, 1.11) |
| **Age (10 years)** | 1.34(1.08, 1.65)** | 1.40(1.10, 1.78)* | 1.31(1.05, 1.62)* | 1.38(1.08, 1.76)* |
| **Marital status** | $\chi^2 = 4.78$ | $\chi^2 = 5.70$ | $\chi^2 = 4.81$ | $\chi^2 = 5.64$ |
| Married | 0.49(0.25, 0.96) | 0.43(0.20, 0.91)* | 0.49(0.24, 0.96) | 0.43(0.20, 0.91)* |
| WDS | 0.59(0.28, 1.27) | 0.57(0.25, 1.32) | 0.60(0.28, 1.29) | 0.57(0.25, 1.32) |
| **Education** | $\chi^2 = 3.80$ | $\chi^2 = 2.69$ | $\chi^2 = 2.40$ | $\chi^2 = 2.02$ |
| Primary | 0.49(0.23, 1.07) | 0.55(0.24, 1.31) | 0.54(0.24, 1.21) | 0.59(0.24, 1.43) |
| Secondary | 0.49(0.20, 1.17) | 0.62(0.23, 1.64) | 0.57(0.22, 1.43) | 0.68(0.24, 1.94) |
| Bachelor+ | 0.42(0.16, 1.11) | 0.45(0.15, 1.33) | 0.51(0.18, 1.44) | 0.51(0.16, 1.67) |
| **Monthly income** | $\chi^2 = 6.49$ | $\chi^2 = 9.67^*$ | $\chi^2 = 6.27$ | $\chi^2 = 9.44$ |
| 5–9.99K (151-300USD) | 1.05(0.59, 1.85) | 0.83(0.44, 1.58) | 1.05(0.59, 1.86) | 0.83(0.44, 1.58) |
| 10–14.9K (301-450USD) | 0.56(0.30, 1.03) | 0.42(0.21, 0.84)* | 0.57(0.31, 1.05) | 0.43(0.21, 0.86) |
| 15–24.9K (451-750USD) | 0.59(0.32, 1.09) | 0.44(0.22, 0.90)* | 0.61(0.33, 1.12) | 0.45(0.22, 0.91) |
| ≥25K (>750USD) | 0.94(0.47, 1.90) | 0.82(0.37, 1.83) | 0.96(0.49, 1.99) | 0.85(0.38, 1.90) |
| **Province (KK)** | 0.50(0.30, 0.83)** | 0.46(0.26, 0.82)* | 0.48(0.29, 0.80)** | 0.45(0.25, 0.80)** |
| **T2D duration (5years)** | 0.91(0.80, 1.03) | 0.93(0.80, 1.08) | 0.91(0.80, 1.03) | 0.93(0.80, 1.08) |
| **T2D treatment** | $\chi^2 = 66.97^{***}$ | $\chi^2 = 20.10^{***}$ | $\chi^2 = 65.77^{***}$ | $\chi^2 = 19.79^{***}$ |
| OHA | 0.51(0.11, 2.35) | 0.89(0.16, 4.98) | 0.54(0.12, 2.45) | 0.91(0.16, 5.13) |
| Insulin | 0.11(0.02, 0.52)** | 0.30(0.05, 1.81) | 0.11(0.02, 0.55)** | 0.31(0.05, 1.86) |
| OHA+Insulin | 0.11(0.02, 0.53)** | 0.34(0.06, 1.98) | 0.12(0.03, 0.56)** | 0.35(0.06, 2.06) |
| **Smoking** | $\chi^2 = 4.782$ | $\chi^2 = 5.413$ | $\chi^2 = 4.715$ | $\chi^2 = 5.244$ |
| Previous | 1.78(0.72, 4.39) | 1.93(0.70, 5.31) | 1.79(0.72, 4.43) | 1.93(0.70, 5.28) |
| Current | 3.34(1.05, 10.62)* | 3.91(1.16, 13.20)* | 3.29(0.04, 10.41) | 3.82(1.13, 12.93) * |
| **Alcohol** | $\chi^2 = 0.998$ | $\chi^2 = 0.112$ | $\chi^2 = 1.07$ | $\chi^2 = 0.121$ |
| Previous | 0.63(0.25, 1.57) | 0.84(0.30, 2.35) | 0.61(0.24, 1.55) | 0.84(0.30, 2.34) |
| Current | 0.81(0.34, 1.91) | 0.96(0.36, 2.59) | 0.81(0.34, 1.92) | 0.97(0.36, 2.60) |
| **BMI** | $\chi^2 = 2.883$ | $\chi^2 = 6.703$ | $\chi^2 = 3.029$ | $\chi^2 = 6.819$ |
| <18.5 | 0.57(0.18, 1.80) | 0.40(0.12, 1.41) | 0.55(0.18, 1.74) | 0.39(0.11, 1.39) |
| 25–29.9 | 0.98(0.64, 1.52) | 0.97(0.59, 1.57) | 0.96(0.62, 1.49) | 0.96(0.58, 1.56) |
| 30+ | 1.33(0.82, 2.15) | 1.62(0.94, 2.78) | 1.31(0.81, 2.13) | 1.61(0.93, 2.77) |
| **Comorbidity** | 1.35(0.62, 2.91) | 1.57(0.64, 3.88) | 1.36(0.63, 2.94) | 1.58(0.64, 3.89) |
| **Family history** | 1.02(0.70, 1.50) | 1.34(0.87, 2.07) | 1.05(0.71, 1.54) | 1.36(0.87, 2.09) |
| **Hospital size** | 0.97(0.62, 1.51) | 1.03(0.63, 1.69) | 0.93(0.60, 1.46) | 1.01(0.61, 1.66) |

Note: χ2 = Likelihood ratio test;

***p<0.001,

**p<0.01,

*p<0.05

Model 1 Effect of Diabetes self-management adjusted only for patient characteristics,

Model 2a Effect of Diabetes self-management adjusted for diabetes management self-efficacy and patient characteristics,

Model 2b Effect of Diabetes self-management adjusted for diabetes knowledge and patient characteristics, and

Model 3 Effect of Diabetes self-management adjusted for both diabetes management self-efficacy, diabetes knowledge along with patient characteristics

All four models are adjusted for all other patient characteristics.

Bangkok patients), and those on insulin (relative to those on no treatment) all had poorer glycaemic control (all p<0.05).

Table 3 provides the adjusted associations between glycaemic control and diabetes self-management (DSM), diabetes management self-efficacy, diabetes knowledge and patient characteristics. However, to further tease out the interplay between diabetes self-management, diabetes management self-efficacy and diabetes knowledge, after adjusting for patient characteristics, we fit four different multivariable models: Model 1 assessed association between HbA1c control and DSM, adjusted only for patient characteristics; Model 2a evaluated the association between HbA1c control and DSM adjusted for DMSE and patient characteristics; Model 2b evaluated the association between HbA1c control and DSM adjusted for DK and patient characteristics; and Model 3 examined the association between HbA1c and DSM adjusted for both DMSE and DK along with patient characteristics.

The results of the multivariable modelling (Table 3) show a strong association was observed between diabetes self-management and glycemic control after adjusting for patient characteristics with each additional unit of SDSCA10 being associated with 1.69 times the odds of HbA1c control (Model 1: OR = 1.69; 95%CI: 1.49, 1.91; p<0.001). The association between diabetes self-management and glycemic control was consistent after additionally adjusting (only) for diabetes knowledge (Model 2b: OR = 1.69; 95%CI: 1.50, 1.91; p<0.001). However, the effect of diabetes self-management on glycemic control was much attenuated after adjusting for diabetes management self-efficacy (Model 2a, Model 3), such that the effect was no longer statistically significant (Model 2a: OR = 1.11; 95%CI: 0.95, 1.29; p = 0.187; Model 3: OR = 1.11; 95%CI: 0.95, 1.29; p = 0.176). A strong positive association was observed between DMSE and glycemic control independent of both self-management and diabetes knowledge (Model 3: OR = 2.67; 95%CI: 2.20, 3.25; p<0.001). Diabetes knowledge was not significantly associated with glycemic control in any of the multivariable models and nor was there any evidence to suggest it confounds the relationships between self-management or diabetes management self-efficacy with glycemic control.

## Discussion

In this study we examine the associations of diabetes self-management, diabetes management self-efficacy and diabetes knowledge with glycemic control. We considered Thai people with type 2 diabetes visiting outpatient departments in either small community or large university hospitals from two provinces (Bangkok and Khon Kaen). Over half our sample failed to achieve glycemic control measured by HbA1c and this result is consistent with studies of other type 2 diabetes populations [12, 38]. Glycemic control has been shown to be one of the most important clinical targets for people with Type 2 diabetes, and the control of blood glucose has been shown to be protective against the development of both acute and chronic complications in Thai [20, 39] and several other populations [12, 40]. To the best of our knowledge, the present study represents one of first to consider the impact of diabetes self-management, diabetes management self-efficacy and diabetes knowledge on blood glucose control among Thai people with type 2 diabetes, and one of only a few to consider all three measures in terms of glycemic control, in any diabetes population.

The results of our bivariate analysis suggest that diabetes self-management, management self-efficacy and knowledge are all associated with glycemic control among Thais with T2D. Somewhat counter-intuitively, higher diabetes knowledge was revealed to be moderately negatively associated with glycemic control in this population. However, when we adjusted for T2D management, management self-efficacy and other patient characteristic we found no evidence to suggest diabetes knowledge is associated with blood glucose control, a result

consistent with that found by Coates [41]. However, this lack of association is inconsistent with several studies considering the impact of diabetes knowledge on glycaemic control. For example, others have demonstrated that higher knowledge is associated with higher control in other populations [16, 17, 42], whereas McPherson [43] showed that diabetes knowledge was inversely associated with glycemic control. This inconsistent result across studies and populations may be for a number of reasons. For example, negative associations between diabetes knowledge and glycemic control may be due to reverse causation; those who have had T2D for longer are likely to have poorer blood glucose control as the disease progresses, but they may have better knowledge of their disease. With the cross-sectional design we employed, we were unable to assess the temporality of the observed associations. Regardless, the lack of association between diabetes knowledge and glycaemic control among Thai people with T2D means we may need to reconsider early educational interventions in Thais with T2D which have a strong emphasis of knowledge transference [44], and typically do not have components which directly focus improving patients' attitude or self-care behaviour. Indeed, a study of the impact of an intervention in Hong Kong people with diabetes demonstrated considerably better post-intervention self-care for those participants who also completed the 'empowerment' component of the program compared to those that did the educational component alone [45].

Our finding that T2D management self-efficacy is strongly associated with glycemic control above and beyond diabetes knowledge and diabetes self-management is an important finding for a number of reasons. Current theory is that disease knowledge and management self-efficacy are both antecedents of disease self-management, and it is only through their impact on disease self-management that they can improve disease outcomes or clinical targets [46, 47]. The results of the present study suggest that diabetes management self-efficacy is not only strongly associated with blood glucose control, but that this association remains even after adjusting for diabetes self-management. There are a number of possible explanations for this. First, the SDSCA scale may be an imperfect measure of diabetes self-management in the Thai T2Dpopulation, and DMSES may be capturing aspects of diabetes management not accounted for by the SDSCA. A second explanation might be reverse causality; achievement of blood glucose control may reinforce the individual's confidence in reaching their clinical targets, and failure to control may diminish one's belief in their ability to control their own disease. Unfortunately, the cross-sectional nature of our study design makes it difficult to explore the reasons why diabetes management self-efficacy remains associated with glycemic control beyond its contribution to diabetes self-management. However, it does represents an interesting avenue of inquiry, and suggests that we don't yet fully understand the complex interplay between disease knowledge, attitude and practice in people with Type 2 diabetes.

Regardless of the apparent strong mediational effect diabetes management self-efficacy had on diabetes self-management in our study, diabetes self-management is still an important predictor of the blood glucose control clinical target. Our results simply suggest that those patients that have a strong belief in their ability to manage their condition, in actuality, are effective at managing their own disease. The more important question is that given diabetes management self-efficacy strong influence on patients' self-care, is diabetes management self-efficacy something that might be amenable to modification through intervention, or is it a more latent trait that might be a spill-over from a patient's general self-efficacy. This is a question that has been investigated, albeit indirectly, in a study of the mechanisms underpinning patient diabetes self-care behaviour, and whether a Patient-Empowerment Programme (an intervention addressing knowledge, attitude and practice among people with T2D) may improve self-care behaviour in people with diabetes in Hong Kong [48]. Indeed the PEP was subsequently shown to improve clinical targets and processes of care [45], longer term patient outcomes including mortality [49] and health-related quality of life [50] in this population. However, the complex interplay

between Diabetes knowledge, management self-efficacy and self-management still needs further investigation. If indeed a patient's diabetes management self-efficacy can be improved than this is likely to represent a cost-effective approach to enhancing patients' self-management thereafter leading to better glycemic control, and ultimately prolonging the onset of chronic diabetes complications.

Our study had several limitations. Perhaps the most important of these was the cross-sectional design we employed. In terms of the strong association between HbA1c control and DMSES we observed, this may mean strong self-efficacy leads to better blood glucose control, but also that successful achievement of blood glucose control may reinforce patients' confidence in their ability to manage their disease. A second limitation is that we only sampled patients from specialized diabetes and general medical clinics. While these types of clinics care for a large majority of T2D outpatients in Thailand there are other avenues of diabetes outpatient care in Thailand. Finally, we only sampled patients from two of Thailand's five regions. However, these regions do represent the best (central region) and worst (North-east region) in terms of diabetes prevalence [20], so by design, we sampled patients from these two regions. Regardless, whether our findings are generalizable to all Thais with T2D is not clear.

The present study also has some strengths. We conducted a rigorous psychometric validation of the three instruments we employed in this study. In addition our response rate was high, with all patients who provided consent, completing the questionnaire. Finally, a majority of studies that have considered the psychometric aspects of diabetes are typically small (n<250) and involve patients attending a single clinic. In contrast, the present study was a moderately large, multicentre study sampling patients from several health-care settings models in Thailand, and from two different regions.

The prevalence of Type 2 Diabetes in developing countries is increasing at an alarming rate. The need for cost effective strategies to minimize diabetes hospitalization, morbidity and mortality is paramount. In this study we evaluated the effect of diabetes knowledge, diabetes management self-efficacy and diabetes self-management of blood glucose control on Thai type 2 diabetes outpatients and found that both diabetes management self-efficacy and diabetes self-management have a substantial effect on patients achieving this clinical target. Our results suggest that diabetes management self-efficacy, something largely ignored in current early educational interventions, is highly associated with blood glucose control. However, the question remains how to best design and implement interventions that can enhance patients' diabetes management self-efficacy.

## Supporting information

**S1 File. English language version of the questionnaire.**
(PDF)

**S2 File. Thai language version of the questionnaire.**
(PDF)

## Author Contributions

**Conceptualization:** Cameron P. Hurst.

**Formal analysis:** Cameron P. Hurst, Karen Hay.

**Investigation:** Cameron P. Hurst, Nitchamon Rakkapao.

**Methodology:** Cameron P. Hurst, Nitchamon Rakkapao, Karen Hay.

**Supervision:** Cameron P. Hurst, Nitchamon Rakkapao.

**Writing – original draft:** Cameron P. Hurst.

**Writing – review & editing:** Nitchamon Rakkapao, Karen Hay.

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
