## [Decision Letter · Decision Letter 0]

15 Oct 2020

PONE-D-20-18140

Impact of diabetes self-management, diabetes management self-efficacy and diabetes knowledge on glycemic control in Type 2 Diabetes (T2D) patients: A prospective multi-center study in Thailand.

PLOS ONE

Dear Dr. Rakkapao,

Thank you for submitting your manuscript to PLOS ONE. After careful consideration, we feel that it has merit but does not fully meet PLOS ONE’s publication criteria as it currently stands. Therefore, we invite you to submit a revised version of the manuscript that addresses the points raised during the review process.

The manuscript has been reviewed by two experts in the filed, and I encourage the authors take full consideration in all these comments as I believe that this will help the authors improve their work.

I am not going to repeat what the reviewers commented; however, I would like the authors to acknowledge that Hong Kong has implemented a Patient Empowerment Programme (PEP) on people with type 2 diabetes. The PEP is based on the concept of self-efficacy and has been found to be effective on self-care behaviors and health outcomes among people with type 2 diabetes. Therefore, the authors should discuss the PEP in their paper.

Please refer to the following references

Wong CK, Wong WC, Lam CL, Wan YF, Wong WH, Chung KL, Dai D, Tsui EL, Fong DY. Effects of Patient Empowerment Programme (PEP) on clinical outcomes and health service utilization in type 2 diabetes mellitus in primary care: an observational matched cohort study. PLoS One. 2014;9(5):e95328.

Lin CY, Cheung MKT, Hung ATF, Poon PKK, Chan SCC, Chan CCH. Can a modified theory of planned behavior explain the effects of empowerment education for people with type 2 diabetes? Ther Adv Endocrinol Metab. 2020;11:2042018819897522.

Wong CK, Wong WC, Wan YF, Chan AK, Chung KL, Chan FW, Lam CL. Patient Empowerment Programme in primary care reduced all-cause mortality and cardiovascular diseases in patients with type 2 diabetes mellitus: a population-based propensity-matched cohort study. Diabetes Obes Metab. 2015;17(2):128-35.

Wong CKH, Wong WCW, Wan EYF, Wong WHT, Chan FWK, Lam CLK. Increased number of structured diabetes education attendance was not associated with the improvement in patient-reported health-related quality of life: results from Patient Empowerment Programme (PEP). Health and Quality of Life Outcomes. 2015;13:126. doi:10.1186/s12955-015-0324-3.

We look forward to receiving your revised manuscript.

Kind regards,

Chung-Ying Lin

Academic Editor

PLOS ONE

Journal Requirements:

2. In your Methods section, please provide additional information about the participant recruitment method and the demographic details of your participants. Please ensure you have provided sufficient details to replicate the analyses such as: a) the recruitment date range (month and year), b) a description of any inclusion/exclusion criteria that were applied to participant recruitment, c) a description of how participants were recruited, and d) descriptions of the specific locations where participants were recruited and where the research took place.

3. Please provide a sample size and power calculation in the Methods, or discuss the reasons for not performing one before study initiation.

4. Please include additional information regarding the survey or questionnaire used in the study and ensure that you have provided sufficient details that others could replicate the analyses. For instance, if you developed a questionnaire as part of this study and it is not under a copyright more restrictive than CC-BY, please include a copy, in both the original language and English, as Supporting Information.

5.We note that you have indicated that data from this study are available upon request. PLOS only allows data to be available upon request if there are legal or ethical restrictions on sharing data publicly. For information on unacceptable data access restrictions, please see http://journals.plos.org/plosone/s/data-availability#loc-unacceptable-data-access-restrictions.

Reviewers' comments:

Reviewer's Responses to Questions

**Comments to the Author**

1. Is the manuscript technically sound, and do the data support the conclusions?

Reviewer #1: Yes

Reviewer #2: Partly

2. Has the statistical analysis been performed appropriately and rigorously? 

Reviewer #1: I Don't Know

Reviewer #2: Yes

3. Have the authors made all data underlying the findings in their manuscript fully available?

Reviewer #1: No

Reviewer #2: No

4. Is the manuscript presented in an intelligible fashion and written in standard English?

Reviewer #1: Yes

Reviewer #2: Yes

5. Review Comments to the Author

Reviewer #1: This is a cross-sectional study of diabetes control in patients in Thailand and relationship to diabetes knowledge, diabetes management self-efficacy (DMSE) and diabetes self-management.

The authors found that diabetes knowledge does not correlate so well with diabetes control as diabetes management self-efficacy. This is important as in Thailand the emphasis is on knowledge and not DMSE.

This is a well-written manuscript, however, there are a few issues that must be resolved.

This is NOT a prospective study- please change the title.

Methods are not complete.

How were the subjects recruited? Were they approached when they were waiting for an office visit? After the visit? Were the subjects approached in their homes? By mail? By phone?

How many of the subjects approached agreed to participate?

What are the inclusion and exclusion criteria?

Did you have to not include any subjects? What were the reasons the subjects were not eligible?

What is meant by the patients were recruited until the study was completed?

If the de-identified data are available by request from author NR, why aren’t these data freely available without restrictions?

Minor comments:

Lines 67-70 states that the increasing prevalence of T2DM is mainly due to an ageing population. Reference #2 does not say that. I do not think that is the cause of increasing prevalence. If there is a reference to support that statement, please provide.

Lines #105-109- please remove “for example” next to your references.

Reviewer #2: Overall, this is a well-done study and well-written paper. There are a few concerns that should be addressed. Specifically:

1. Data availability- As I read the journal instructions, having the data available through contact with an author is not an acceptable alternative. The authors explain that the original dataset has identifiable information, so it is OK to restrict access to that. But the authors worked with a deidentified dataset, so I don’t think that author contact is acceptable. But this is a point for the journal editors to address.

2. The Abstract should have a sentence or two describing the reasons for doing this study and predetermined hypotheses.

3. Line 68- While aging of the population certainly contributes to increasing prevalence, I don’t think you can say that is the “main” cause- rising rates of obesity and sedentary lifestyle are also factors (which I see that you note in the following paragraph).

4. Line 96- Most medical journals use the term “blood glucose,” not “blood sugar,” which is a more colloquial term. Similarly, most diabetes journals no longer accept “diabetes patient,” as it is deemed to be stigmatizing. Instead one writes “person with diabetes,” some even shorten that to “PWD.” These are minor quibbles, but making these changes throughout the manuscript would conform to currently accepted terminology.

5. Please say a little more about recruitment procedures, i.e., how were potential participants approached. Of those approached, how many refused? Whle the fact that those who consented did complete the questionnaires is good, the reader also needs to know something about those who refused, to address generalizabiity of the results. Also, you state that data were gathered until it was complete, meaning that you had 700 participants. How did you determine that you needed 700 participants?

6. Line 135- It’s Toobert, not Toober.

7. Very nice work validating the measures for use in a sample of Thai individuals.

8. Why did you choose to look at A1c as a dichotomous variable? You lose a lot of data points by doing so, as we know that an A1c of 7.0% isn’t very different than one of 7.1%, yet the former is a failure and the latter is a success. Please either perform longitudinal analyses, or explain why a dichotomous approach is appropriate.

9. Table 2- Suggest you create a simpler title, and put info about the different models as footnotes- very hard to read.

10. Line 324- The statement that interventions should not focus on diabetes knowledge, since knowledge didn’t relate to glycemic control, is an over-reach. Given the cross-sectional nature of the data, you might find that knowledge does predict A1c longitudinally. Also, there are other benefits to knowledge, and I’d assume that high knowledge relates to high self-efficacy. I may have missed it, but did you look at the relationships between the measures? If not, you should, as it might clarify what you’ve found.

11. The diabetes self-efficacy discussion is a little convoluted. It certainly makes sense that people who manage their diabetes well also have a high level of self-efficacy, i.e., they believe they can do it because they are doing it (or they are doing it because they believe they can do it- who knows?). The notion that you should target self-efficacy, as if it’s separate from actual self-management has always troubled, and confused, me. I don’t think self-efficacy is “largely ignored” in interventions, I think it’s just assumed that improving knowledge, and improving self-care, will result in improving self-efficacy too. I’m not sure what changes I’d recommend, but these conclusions need attention. Perhaps future research needs to take a more longitudinal approach to test out the hypotheses I’ve stated.

6. PLOS authors have the option to publish the peer review history of their article (what does this mean?). If published, this will include your full peer review and any attached files.

Reviewer #1: No

Reviewer #2: No

---

## [Author Response · Author response to Decision Letter 0]

13 Dec 2020

Please see the attached document 'Response to reviewers.doc' for details of our response to each individual comment

---

## [Editor Report · Decision Letter 1]

15 Dec 2020

Impact of diabetes self-management, diabetes management self-efficacy and diabetes knowledge on glycemic control in Type 2 Diabetes (T2D) patients: A prospective multi-center study in Thailand.

PONE-D-20-18140R1

Dear Dr. Rakkapao,

We’re pleased to inform you that your manuscript has been judged scientifically suitable for publication and will be formally accepted for publication once it meets all outstanding technical requirements.

Kind regards,

Chung-Ying Lin

Academic Editor

PLOS ONE

Additional Editor Comments (optional):

I thank the authors satisfactorily responded to all the reviewers' and mine comments. I have read through the responses and the revised manuscript; I found that the revised manuscript is acceptable. Just one minor issue. It seems that the authors took out the word "prospective" in their submitted word file but did not do so in the system. Please make sure to fully take out the word during proof stage.
---

## [Editor Report · Acceptance letter]

18 Dec 2020

PONE-D-20-18140R1 

Impact of diabetes self-management, diabetes management self-efficacy and diabetes knowledge on glycemic control in people with Type 2 Diabetes (T2D): A multi-center study in Thailand. 

Dear Dr. Rakkapao:

I'm pleased to inform you that your manuscript has been deemed suitable for publication in PLOS ONE. Congratulations! Your manuscript is now with our production department. 

Kind regards, 

on behalf of

Dr. Chung-Ying Lin 

Academic Editor

PLOS ONE